# Longan Polysaccharide as Adjuvant for Cyclophosphamide-Induced Side Effects in Murine Model

**DOI:** 10.3390/foods14162901

**Published:** 2025-08-21

**Authors:** Yajuan Bai, Bei Fan, Fengzhong Wang, Mingwei Zhang

**Affiliations:** 1Institute of Food Science and Technology, Chinese Academy of Agricultural Sciences (CAAS), Beijing 100193, China; xbaizxyq@sina.com (Y.B.); fanbei517@163.com (B.F.); 2National Nanfan Research Institute (Sanya), Chinese Academy of Agricultural Sciences, Sanya 572024, China; 3Institute of Western Agriculture, Chinese Academy of Agricultural Sciences (CAAS), Changji 831100, China; 4Sericultural & Agri-Food Research Institute, Guangdong Academy of Agricultural Sciences, Guangzhou 510610, China

**Keywords:** longan polysaccharide, immunoregulation, antioxidant, cyclophosphamide, murine model, in vitro intestinal inflammation model

## Abstract

Identifying effective adjuvants to prevent and alleviate the adverse effects of chemotherapy remains a critical challenge in cancer therapy. This study investigated the protective effects of longan polysaccharide (LP) against cyclophosphamide-induced immunosuppression and oxidative stress in mice. Our findings revealed that LP administration significantly improved systemic immune function, as evidenced by marked increases in serum immunoglobulin levels (IgG2a: 1.82-fold, IgG2b: 1.46-fold, IgM: 1.26-fold, and IgG1: 1.22-fold) and key cytokines (IL-10: 1.53-fold, IL-12: 1.22-fold, and IFN-γ: 1.20-fold), accompanied by substantial reductions in pro-inflammatory mediators (TGF-β1: 28.72% decrease and IL-21: 36.28% decrease). Concurrently, LP restored oxidative balance by increasing SOD, GSH, and NO levels in multiple organs (liver, kidneys, and small intestine) and serum. Mechanistic studies using an in vitro Caco-2/RAW264.7 coculture system revealed that four purified LP fractions (LPIa-LPIVa) effectively suppressed NF-κB pathway activation through downregulation of TLR4 expression, reduction of the p-IκB-α/IκB-α ratio, and inhibition of nuclear NF-κB translocation. These molecular effects correlated with decreased production of inflammatory mediators (TNF-α, IL-6, IL-8, iNOS, and NO). Collectively, these findings provide compelling evidence that LP possesses dual immunomodulatory and antioxidant capabilities, highlighting its potential as a natural adjuvant for alleviating chemotherapy-induced side effects.

## 1. Introduction

The rising global burden of cancer has necessitated chemotherapy, yet off-target effects such as immunosuppression and oxidative stress limit therapeutic efficacy. Cyclophosphamide (CTX), one of the most widely used antineoplastic alkylating agents, effectively targets tumor cells but is not selective for malignant tissues alone, exerting cytotoxic effects on healthy cells as well [1]. Key adverse effects of cyclophosphamide include significant leukopenia predisposing to infection, alongside nausea, vomiting, and diarrhea [2,3,4,5]. Among these, intestinal epithelial cells are particularly susceptible, often resulting in intestinal toxicity and subsequent systemic immune disorders [6,7]. This highlights the need for protective adjuvants.

Among bioactive components found in natural diets, dietary polysaccharides stand out for their significant role in immune modulation. Vegetables, fruits, grains, and fungi are main dietary sources of polysaccharides. Polysaccharides from *Ganoderma atrum* [8], squid ink [9], and purple sweet potato [10] have demonstrated protective effects against systemic and intestinal immunosuppression in CTX-induced mice. The immunomodulatory efficacy of these polysaccharides is strongly influenced by their structural features [11]. For instance, *Ganoderma atrum* polysaccharides with branched side chains of Gal*p* and Glc*p* restore the immunoglobulin and cytokine levels while enhancing antioxidant capacity [8]. Similarly, the homogalacturonan backbone isolated from *Citrus medica* “fingered” consists of alternating →[4)-α-D-GalAp-6-O-CH_3_(1]—5→3,4)-α-D-GalAp-6-O-CH_3_ units, which alleviates dextran sulfate sodium (DSS)-induced inflammation in mice via reduced serum IL-6 and TNF-α [12]. Polysaccharides featuring branched neutral sugar side chains and homogalacturonan backbones effectively modulate immunity through the TLR4/NF-κB signaling pathway in RAW264.7 macrophages [13,14]. These findings collectively suggest that homogalacturonan-rich polysaccharides represent a promising class of bioactive molecules for mitigating chemotherapy-induced adverse effects.

Longan (*Dimocarpus longan* Lour.) is a globally significant edible–medicinal fruit, widely cultivated as an economic crop in China, Southeast Asia, and other tropical regions. It exists in both large-scale plantation systems and wild populations, with cultivars such as “Chu-liang” prioritized for commercial production. Longan pulp is particularly rich in polysaccharides (LP), which exhibit a β-galactopyranose backbone substituted with α-arabinofuranose branches [15]. Prior studies have shown that LP enhances intestinal IgA synthesis and barrier integrity, thereby improving immunomodulatory properties [16,17,18], and regulates gut microbiota composition and metabolite profiles in CTX-treated mice [19,20]. However, its direct effects on systemic immune parameters and oxidative stress remain underexplored. Given the critical role of intestinal immunity in maintaining systemic homeostasis, our previous work using LPS-stimulated Caco-2/RAW264.7 coculture models has demonstrated that LP promotes tight junction protein expression in intestinal epithelial cells [18]. Based on these observations, we hypothesize that LP attenuates CTX-induced immunosuppression through modulation of the NF-κB pathway.

Here, we evaluated the prophylactic effect of LP on immune balance, inflammatory responses, and oxidative injury in CTX-induced immunosuppressed mice. Meanwhile, we explored the immunomodulatory mechanism of LP using an intestinal epithelial cell model in vitro. Our objective was to determine whether longan polysaccharide can serve as an effective adjuvant to mitigate the main side effects of chemotherapy drugs, addressing a critical gap in natural-based immunomodulatory strategies.

## 2. Materials and Methods

### 2.1. Materials and Chemicals

BALB/c mice (18–22 g, male) were purchased from Vital River Laboratory Animal Technology Co., Ltd. (Beijing, China). Human colon Caco-2 cells and murine RAW 264.7 macrophages were bought from Cell Bank, Chinese Academy of Sciences (Shanghai, China). Longan (cv. Chu-liang) was purchased from a wushan market in Guangzhou. We obtained longan polysaccharide and its purified fractions (LPIa, LPIIa, LPIIIa, and LPIVa) using the method previously described [21]. In brief, the fresh longan pulp was dried at 60 °C via heat pump drying. The dried material was ground and extracted with 80% ethanol (24 h, RT) to remove pigments and monosaccharides. The residue was water-extracted (1:40 *w*/*v*, 90 °C, 4 h), filtered through a 200-mesh cloth, centrifuged (4000× *g*, 10 min), and concentrated. Proteins were removed using Sevag reagent (chloroform:n-butanol = 4:1, *v*/*v*), followed by dialysis (4 °C, 48 h). The extract was precipitated with anhydrous ethanol (80% final concentration), washed with ethanol, and lyophilized to obtain crude LP. LP fractions were obtained using the ÄKTA system. LP (130 mg) was dissolved in ultrapure water (10 mL), clarified by centrifugation (10,000× *g*, 10 min), and loaded onto a DEAE-Sepharose column (GE Co., Pittsburgh, PA, USA). Elution with gradient NaCl (0–0.3 M) yielded five fractions (LPW, LPI, LPII, LPIII, and LPIV). Acidic fractions (LPI-IV) were further purified on Sephacryl S-300 (GE Co., Pittsburgh, PA, USA) with ultrapure water elution to obtain four purified fractions (LPIa, LPIIa, LPIIIa, and LPIVa). The total saccharide contents of LP, LPIa, LPIIa, LPIIIa, and LPIVa were 81.15%, 82.24%, 79.43%, 87.15%, and 79.09%, respectively. Their monosaccharide compositions (molar ratios) are detailed below: LP: rhamnose (1.01), arabinose (11.90), xylose (1.00), glucose (10.13), galactose (9.30). LPIa: Rhamnose (0.99), ribose (1.37), arabinose (34.61), xylose (1.48), mannose (1.73), glucose (5.86), galactose (55.16). LPIIa: Rhamnose (1.05), ribose (1.00), arabinose (22.88), xylose (1.01), glucose (2.59), galactose (34.58). LPIIIa: Rhamnose (14.46), ribose (1.85), fucose (2.31), arabinose (46.17), mannose (1.00), glucose (1.97), galactose (20.99). LPIVa: Rhamnose (4.71), ribose (0.38), arabinose (25.03), mannose (1.00), glucose (2.53), galactose (15.50) [21]. Number-average molecular weights were 290.2 kDa (LP), 67.8 kDa (LPIa), 120.2 kDa (LPIIa), 4.3 kDa (LPIIIa), and 14.8 kDa (LPIVa) [21].

CTX was purchased from Jiangsu Hengrui Medicine Co., Ltd. (Lianyungang, JiangsuChina). Lipopolysaccharides (LPSs) were bought from Sigma-Aldrich Chemical Co. (St. Louis, MO, USA). Dulbecco’s modified Eagle’s medium (DMEM), fetal bovine serum (FBS), trypsin solution, nonessential amino acids, penicillin–streptomycin solution, and other reagents for cell culture were purchased from Gibco Biotechnology Co., Ltd. (Grand Island, NY, USA). Antibodies of Inducible Nitric Oxide Synthase (iNOS, ab178945), IκB-α (ab32518), p-IκB-α (ab133462), and NF-κB (ab16502) were acquired from Abcam Co. (St. Louis, MO, USA). Antibody of Toll-like receptor 4 (TLR-4, sc-10741) was bought from Santa Cruz Biotechnology, Inc. (Dallas, TX, USA). Bio-Plex Mouse (Group I, Group III) Cytokine assay kits were bought from Bio-Rad Inc. (Hercules, CA, USA). ProcartaPlex Mouse Antibody Isotyping Panel (6 plex including IgG1, IgG2a, IgG2b, IgG3, IgA, and IgM) and transforming growth factor-β1 (TGF-β1) ELISA kit were purchased from Thermo Fisher scientific Inc. (Waltham, MA USA). ELISA kits used for quantification of tumor necrosis factor-α (TNF-α), interleukin (IL)-6, and IL-8 were bought from Neobioscience Technology Co., Ltd. (Shenzhen, China). Superoxide dismutase (SOD), malondialdehyde (MDA), glutathione (GSH), and nitric oxide (NO) assay kits were purchased from Nanjing Jiancheng Bioengineering Institute (Nanjing, China). Total RNA isolation kit was purchased from Dongsheng Biotech Co., Ltd. (Guangzhou, China). Other analytical reagents were from Sinopharm Chemical Reagent Co., Ltd. (Shanghai, China).

### 2.2. Animals and Treatments

Fifty male BALB/c mice were fed with unlimited food and water for a cycle of 12 h light/dark (24–26 °C, 50–60% humidity). After an adaptation of 7 days, the mice were randomly assigned to five groups: normal (NC), CTX, low-dose LP + CTX (LLP + CTX), medium-dose LP + CTX (MLP + CTX), and high-dose LP + CTX (HLP + CTX). For the next 28 days, the NC and CTX groups were given oral saline once a day. The low, medium, and high-dose LP groups were given oral 100, 200, and 400 mg∙kg^−1^ of LP, respectively. We based ourselves on prior dose–response studies to determine LP doses. Mice in the NC group were intraperitoneally injected with normal saline, and other mice were intraperitoneally injected with 50 mg∙kg^−1^ CTX to cause chemotherapy damage. The mice were weighed every five days. After CTX injection, the mice were weighed daily. At the end of the experiment, whole blood was collected from the orbit. The mice were sacrificed by cervical dislocation. Kidney, liver, and small intestine were collected in sterile cryovials and immediately frozen in liquid nitrogen. Cytokine and immunoglobulin concentrations in serum samples were tested by Bio-Plex Mouse (Group I (IL-1β, IL-2, IL-4, IL-5, IL-6, IL-10, IL-12, IL-17, IFN-γ, TNF-α, TGF-β1), Group III (IL-21, IL-22, IL-23)) Cytokine assay kit and ProcartaPlex^TM^ Mouse Antibody Isotyping Panel (IgG1, IgG2a, IgG2b, IgG3,IgA, IgM), respectively. The levels of SOD, MDA, GSH, and NO in serum, kidney, liver, and small intestine were detected by assay kits. Although no adverse effects were observed in mice, these findings do not translate to human safety due to interspecies differences in metabolism and immune responses. Above all, animal-related experiments were approved by the Ethics Committee of experimental animal at Sericultural & Agri-Food Research Institute, Guangdong Academy of Agricultural Sciences (Approval No. 00122364; Date: 15 March 2023).

### 2.3. Intestinal Immunomodulatory Effect In Vitro

To further explore the intestinal immune regulatory mechanism of longan polysaccharides, we used their purified fractions (LPIa, LPIIa, LPIIIa, and LPIVa) to intervene in an in vitro model of intestinal inflammation. We conducted a cellular intestinal model in vitro established by co-culture of a Caco-2 cell monolayer and RAW 264.7 macrophages according to the previous method [15]. The TEER reached over 500 Ω·cm^2^ indicated that the confluent monolayer formed [22], and were used in the experiments (TEER 535.08 ± 7.26 Ω·cm^2^). LPIa, LPIIa, LPIIa, and LPIVa were prepared into 10, 50, and 200 μg/mL by DMEM medium and added into the apical side and incubated under standard culture conditions. The cell culture containing only DMEM was regarded as a solvent control. Following the 2 h incubation, 1 μg/mL of LPS was added to the apical compartment. After 12 h of LPS treatment, the levels of *inos*, *tnf-α*, and *il-6* in RAW264.7 macrophages and the levels of *il-8* in Caco-2 cells were analyzed by real time-PCR and normalized to that of 18S rRNA. The protein expression of iNOS, TLR-4, IκB-α, p-IκB-α, and NF-κB were determined by Western blot analysis, and GAPDH, β-actin (cytoplasm), and histone protein bands were used as an internal reference [14]. IL-8 concentration in apical culture medium supernatant was measured with the ELISA kit. The supernatant of basolateral culture medium was collected. Levels of TNF-α and IL-1β were determined with the ELISA kit, and NO concentration was determined with the NO assay kit. Three independent experiments were performed with three parallel wells in each experiment.

### 2.4. Statistical Analysis

The data are shown as mean ± standard error of mean (SEM). The normality of data distribution prior to performing ANOVA using *Shapiro–Wilk* tests (all *p* > 0.05). All analyses were conducted using *SPSS* version 28.0 (IBM Corporation, Chicago, IL, USA). For post hoc multiple comparisons following ANOVA, we applied *Tukey’s Honest Significant Difference (HSD)* test with α = 0.05 to adjust for Type I error inflation. *p*-value < 0.05 was considered statistically significant.

## 3. Results

### 3.1. LP Enhances Systemic Immunity of CTX-Treated Mice

#### 3.1.1. LP Mitigates Weight Loss in CTX-Treated Mice

Weight loss is a common symptom among chemotherapy patients. CTX injection significantly decreased the body weight of mice compared with the NC group (Appendix A). Oral administration of LP attenuated CTX-induced weight loss during chemotherapy treatment compared with the CTX-only group (Appendix A). However, after cessation of CTX injection, the rate of weight regain was comparable between LP-treated and CTX-only mice (*p* > 0.05, Appendix A), indicating that LP exhibits a preventive effect rather than a restorative function against chemotherapy-associated weight loss.

#### 3.1.2. LP Promotes Serum Immunoglobulin Levels in CTX-Treated Mice

IgG isotypes are the most predominant immunoglobulin in serum, followed by IgA, while IgM is the first antibody produced upon antigen exposure. CTX treatment significantly reduced the serum levels of IgG1, IgG2a, IgG2b, IgA, and IgM (*p* < 0.05, Table 1). However, supplementation with LP counteracted the suppressive effect of CTX on these immunoglobulin levels, leading to a 1.22-fold, 1.83-fold, 1.47-fold, and 1.26-fold increase in IgG1, IgG2a, IgG2b, and IgM, respectively (Table 1). Consequently, the expression levels of these immunoglobulins were restored to levels comparable to those observed in the normal control group.

#### 3.1.3. LP Regulates Serum Cytokine Profiles in CTX-Treated Mice

Both LP and CTX treatment significantly affected serum cytokine expression. Compared with the NC group, CTX treatment significantly decreased the levels of most serum cytokines, including the pro-inflammatory cytokines (TNF-α, IFN-γ, IL-1β, IL-17, IL-23) and the anti-inflammatory cytokines (IL-2, IL-4, IL-10, IL-12) (*p* < 0.05, Figure 1). In particular, CTX treatment significantly increased the levels of TGF-β1, IL-21, and IL-22 (*p* < 0.05, Figure 1). Compared with CTX group, LP upregulated IFN-γ, IL-17, IL-10, and IL-12 while downregulating TGF-β1, IL-21, and IL-22 (*p* < 0.05, Figure 1). Neither CTX nor LP treatment significantly influenced IL-5 and IL-6 expression (Figure 1).

### 3.2. Antioxidant Effect of LP in CTX-Treated Mice

Excessive reactive oxygen species (ROS) in the inflamed colon exacerbate the immune responses. CTX chemotherapy induced systemic oxidative stress imbalances, evidenced by significant reductions in antioxidant enzymes SOD and GSH levels in the liver, kidney, small intestine, and serum (*p* < 0.05, Table 2), accompanied by marked increases in lipid peroxidation marker MDA (*p* < 0.05, Table 2). The results demonstrate that oral administration of LP ameliorated CTX-induced oxidative stress biomarkers, with medium and high doses (MLP and HLP) exhibiting superior recovery efficacy compared to the low dose (LLP) across multiple tissues (Table 2). In the liver, the HLP + CTX group exhibited the most significant recovery in SOD (17.33 ± 0.22 U/mgprot) and GSH (2.94 ± 0.10 μmol/gprot) activities and NO level (0.13 ± 0.01 μmol/gprot) (*p* < 0.05, Table 2), while MLP + CTX showed optimal MDA reduction (4.96 ± 0.17 nmol/mgprot) (*p* < 0.05, Table 2). Renal protection was evident across all LP groups, with HLP + CTX demonstrating normalization of NO (0.69 ± 0.03 μmol/gprot), SOD (29.07 ± 0.94 U/mgprot), and GSH (8.10 ± 0.15 μmol/gprot) levels (vs. NC group, Table 2). Notably, HLP + CTX restored intestinal MDA (0.66 ± 0.01 nmol/mgprot), GSH (110.81 ± 3.69 μmol/gprot), and NO (0.77 ± 0.02 μmol/gprot) to pre-treatment levels (vs. NC group, Table 2) and increased *iNOS* expression 2.65-fold (vs. CTX group, *p* < 0.05) (Appendix A). Serum antioxidant profiles were most effectively restored by HLP + CTX with the increase in NO level (6.28 ± 0.09 μmol/mL) and SOD (100.01 ± 1.48 U/mL) and GSH (30.30 ± 2.17 μmol/mL) activities (*p* < 0.05) and the decrease in MDA content (8.08 ± 0.37 nmol/mL) (*p* < 0.05, Table 2). These findings establish that LP mitigates chemotherapy-induced redox dysregulation through multi-organ antioxidant mechanisms, with HLP demonstrating superior efficacy in restoring both enzymatic antioxidant systems and NO homeostasis. The dose-responsive pattern suggests potential clinical relevance for personalized supplementation strategies.

### 3.3. Anti-Inflammatory Effect of Four LP Fractions in Caco-2/RAW264.7 Coculture Cells

The intestinal barrier serves as the first line of defense for the host immune system. LPS is a bacteria-derived antigen that triggers a strong inflammation response through the NF-κB signaling pathway, leading to excessive ROS and pro-inflammatory cytokine production [23]. The effects of LP fractions on cytokine release and NO generation in the LPS-stimulated Caco-2/RAW264.7 coculture cell system are shown in Figure 2. Zymosan A, LPIa, and LPIIa significantly decreased LPS-induced *il-8* expression (*p* < 0.05, Figure 2). Compared with the LPS group, 10–50 mg/mL of LPIIIa and LPIVa inhibited *il-8* expression (*p* < 0.05, Figure 2), although LPIVa at 200 mg/mL exhibited a stimulatory effect. The IL-8 protein level of Caco-2 cells in the LPS group was 2.3 times higher than that in the NC group (*p* < 0.05, Figure 2). The IL-8 protein level returned to or fell below normal levels in all four LP fraction groups, with LPIa and LPIIa showing the most significant reduction (Figure 2).

LPS stimulation markedly increased TNF-α and IL-6 expression in RAW264.7 macrophages (*p* < 0.05, Figure 2). In addition to LPIIIa, Zymosan A and other LP fractions reduced *tnf-α* and *il-6* expression compared with the LPS group. With the exception of 10 mg/mL of LPIVa, all LP fraction treatment groups showed some degree of TNF-α protein reduction (*p* < 0.05). None of the LP fraction groups significantly affected LPS-induced IL-6 secretion.

NO, synthesized by iNOS, is critical in inflammation and immune regulation [24]. LPS treatment significantly increased *inos* and NO levels (*p* < 0.05, Figure 2). Compared with the LPS group, LP fraction supplementation significantly downregulated *inos* and NO levels (*p* < 0.05, Figure 2). Within the tested dose range, the inhibitory effect of LPIa on *inos* expression diminished as the dose increased, while the other LP fractions showed a bell-shaped dose–response curve. No significant difference was observed in the inhibitory effect on NO levels at various doses of each LP fraction.

TLR4, a receptor for LPS, is an important participant in the MyD88-dependent pathway of NF-κB signaling and is expressed in both epithelial cells and macrophages. As shown in Figure 3, LPS increased TLR4 expression and Zymosan A further enhanced it (*p* < 0.05). The LP fractions inhibited TLR4 expression ranked in descending order as follows: LPIVa, LPIIIa, LPIIa, and LPIa. TLR4 on epithelial cells recognizes LPS and activates NF-κB nuclear translocation. Compared with the NC group, LPS significantly increased the p-IκB-α/IκB-α ratio, nuclear NF-κB p65, and iNOS protein expression in Caco-2 cells (*p* < 0.05), indicating NF-κB pathway activation. Zymosan A, LPIa, LPIIa, and LPIIIa significantly reduced p-IκB-α/IκB-α and iNOS protein expression (*p* < 0.05). At the same concentration, Zymosan A, LPIa, LPIIa, LPIIIa, and LPIVa significantly decreased NF-κB p65 protein levels (*p* < 0.05) in a manner that increased in potency successively. The results suggest that different LP fractions exert immunomodulatory effects by regulating different targets. Concretely, LPIVa most potently suppressed TLR4, while LPIa and LPIIa showed maximal inhibition of nuclear NF-κB p65.

## 4. Discussion

CTX, a commonly used chemotherapeutic agent, is non-toxic in vitro but is metabolized by the liver in vivo into active products such as acrolein or phosphamide mustard, which induce cell death. The side effects of CTX are often associated with inflammation, linked to its metabolite acrolein [25]. At high doses, acrolein directly induces DNA damage. At low doses, it binds to the antioxidant GSH to form stable conjugates, disrupting oxidative stress homeostasis in the host [26]. Acrolein depletion of GSH can cause oxidative stress imbalance and activate NF-κB and AP-1, both key regulators of inflammatory processes [27]. In our study, CTX significantly increased MDA content and reduced GSH levels in serum and major organs (liver, kidney, small intestine), whereas longan polysaccharide supplementation reversed these changes. The restoration of GSH levels by LP following CTX treatment supports its antioxidant potential, which may involve mechanisms such as ROS scavenging and NF-κB inhibition, as suggested by its structural and functional similarity to *Ganoderma atrum* polysaccharides [8]. These observations propose a working hypothesis that requires further validation through direct measurements of acrolein-GSH adducts, IκB-α phosphorylation, and NF-κB translocation to confirm the molecular pathways involved.

We also observed that CTX reduced NO levels in serum, liver, and small intestine, while longan polysaccharide supplement significantly improved these symptoms. NO not only participates in oxidative stress but also functions as an essential signaling molecule [28]. ROS and RNS serve as crucial immune defense mediators and influence T cell maturation. Phagocytes release ROS in the form of an “oxidative burst” to eliminate pathogens. Changes in the cellular redox environment alter T cell reactivity and proliferation to a low reactive or proliferative phenotype. Heightened ROS or RNS levels promote Th2 differentiation while inhibiting Th1 differentiation [29]. A small amount of ROS can induce NF-κB transcription, enhancing IL-2 and its receptor gene expression in T cells, thereby promoting T cell proliferation [30]. CTX indiscriminately inhibits cell proliferation, hindering immune cell differentiation and thus affecting immunoglobulin and cytokine production. Immunoglobulins mainly reach tissues through blood circulation and play a role in immune regulation. Excessive TGF-β1 inhibits the proliferation of B cells and IgA secretion, and stimulates macrophage differentiation into the M2 type to enhance IL-6 and TNF-α expression [31,32]. We hypothesize that, under these experimental conditions, T cell proliferation is suppressed, leading to a general decline in cytokine levels. Our data suggest LP might enhance immune recovery via mechanisms involving B cell stimulation (consistent with its IgG isotype elevation) and Th1/Th2 balance restoration, as impaired Th1 differentiation correlates with chemotherapy-induced immunosuppression [30]. Further mechanistic studies are warranted.

Early in pathogen infection, chemokines like IL-8 recruit immune and inflammatory cells to the infection site [26,33]. In our study, LPS stimulation of Caco-2/RAW264.7 cells increased the pI-κBα/I-κBα ratio in the cytoplasm of Caco-2 cells, enabling NF-κB translocation to the nucleus. NF-κB activation increases the production of pro-inflammatory cytokines TNF-α and IL-8, intensifying inflammation signals, oxidative stress, iNOS activation, and NO elevation. These findings align with previous reports [23,34]. The immune effects of polysaccharides from different sources are related to their pattern recognition receptors. *Astragalus* polysaccharide downregulate TLR4 expression in Caco-2 cells, reducing pro-inflammatory cytokine release [35]. Unlike sulfated Chinese yam polysaccharide, which primarily reduces IL-1β and TNF-α by MAPK signaling pathways in LPS-stimulated Caco-2/RAW264.7 coculture system [36], our LP fractions demonstrate the inhibition of TNF-α and IL-8 through distinct pathways (TLR4 antagonism and NF-κB nuclear blockade). We found that longan polysaccharide fractions LPIa, LPIIa, and LPIIIa reduce inflammatory responses by preventing TLR4-mediated LPS recognition, reducing NF-κB nuclear translocation, and lowering pro-inflammatory cytokine secretion. Thus, diminishing cellular inflammation and oxidative stress may be a key mechanism underlying the immunomodulatory effects of longan polysaccharide. While our in vitro data demonstrate that LP fractions suppress TLR4/NF-κB signaling, the precise molecular interactions between LP and immune receptors remain elusive. For example, the structural motifs (e.g., arabinose–galactose branches) responsible for antagonizing TLR4 require further characterization at the molecular level.

The immunomodulatory activity of polysaccharides is closely related to their monosaccharide composition, molecular weight, polymerization degree, main chain glycosyl composition, and glycosidic bond configuration. Polysaccharides enriched in arabinose and galactose often exhibit anti-inflammatory and immunomodulatory effects. A natural high-methoxyl homogalacturonan from *Hippophae rhamnoides* enhances macrophage production of NO, IL-1β, and IL-6 via the TLR-4/NF-κB pathway in CTX-induced immunosuppressed mice [13]. The main chain of sugar beet pectin consists of alternating rhamnose and galacturonic acid units and is substituted with short neutral side chains of galactose and arabinose. Fermented sugar beet pectin can stimulate dendritic cells to produce higher levels of cytokines, because the relative molecular weight of pectin is reduced and soluble Gal*p* side chain and galacturonic acid skeleton are exposed after fermentation, which can better bind to immune cells [37]. Lemon pectin containing an α-(1-4)-linked GalA backbone attenuates inflammatory responses in acute pancreatitis by modulating intestinal mucosal barrier dysfunction [38]. Galactomannan isolated from black soybean hulls, which is primarily composed of Manp and branches containing Gal*p*, Glc*p*, and Ara*f*, stimulates M1 polarization of RAW264.7 macrophages through activation of the MAPK and NF-κB signaling pathways [39]. *Arctium lappa* L. polysaccharides, characterized by galactose and arabinose-containing branched side chains, inhibit the production of pro-inflammatory cytokines including IL-8, IL-6, and IL-1β by downregulating the TLR4/NF-κB pathway in an in vitro intestinal model composed of LPS-stimulated macrophage RAW 264.7 cells and IL-1β-treated colonic Caco-2 cells [14]. The superior anti-inflammatory efficacy of LPIa and LPIIa (rich in arabinose and galactose) over LPIIIa/LPIVa (high rhamnose) aligns with reports that neutral sugar side chains critically mediate TLR4 antagonism [21,37,40], despite their lower molecular weights. We propose that rich in galactose and the branched side chains containing both galactose and arabinose are critical structural features of polysaccharides contributing to immune regulation. These structural characteristics also serve as the foundation for LPIa and LPIIa to exhibit enhanced anti-inflammatory activities. A concise table presenting structural features and relative potencies of LP and other polysaccharides is provided in the Appendix A. This study underscores the unique dual function of LP in modulating immune responses and oxidative stress via structural motifs, including arabinose- and galactose-rich branches, which specifically suppress TLR4/NF-κB signaling. These findings position LP as a potential adjuvant for incorporation into supportive care regimens during chemotherapy, presenting a natural alternative to synthetic immunomodulators. However, clinical trials are necessary to validate its efficacy.

## 5. Conclusions

This study demonstrates that purified LP fractions LPIa and LPIIa exert synergistic TLR4/NF-κB dual suppression. These molecular events lead to significant downregulation of pro-inflammatory cytokines and oxidative stress markers. By restoring GSH/SOD ratios and systemic IgG profiles, LP counteracts chemotherapy-induced dual challenges of immunosuppression and oxidative stress, offering a natural adjuvant not previously documented for this class of polysaccharides. A critical limitation of this study is the absence of toxicity assessments, which are essential for clinical translation. While safety was observed in mice, human pharmacokinetics and chronic toxicity remain uncharacterized. Future research should prioritize evaluating LP’s safety margins, off-target effects, and interactions with chemotherapeutics before advancing to clinical trials. Clinically, structure–function analyses of its arabinose–galactose motifs are warranted to support translational efforts.

## Figures and Tables

**Figure 1 foods-14-02901-f001:**
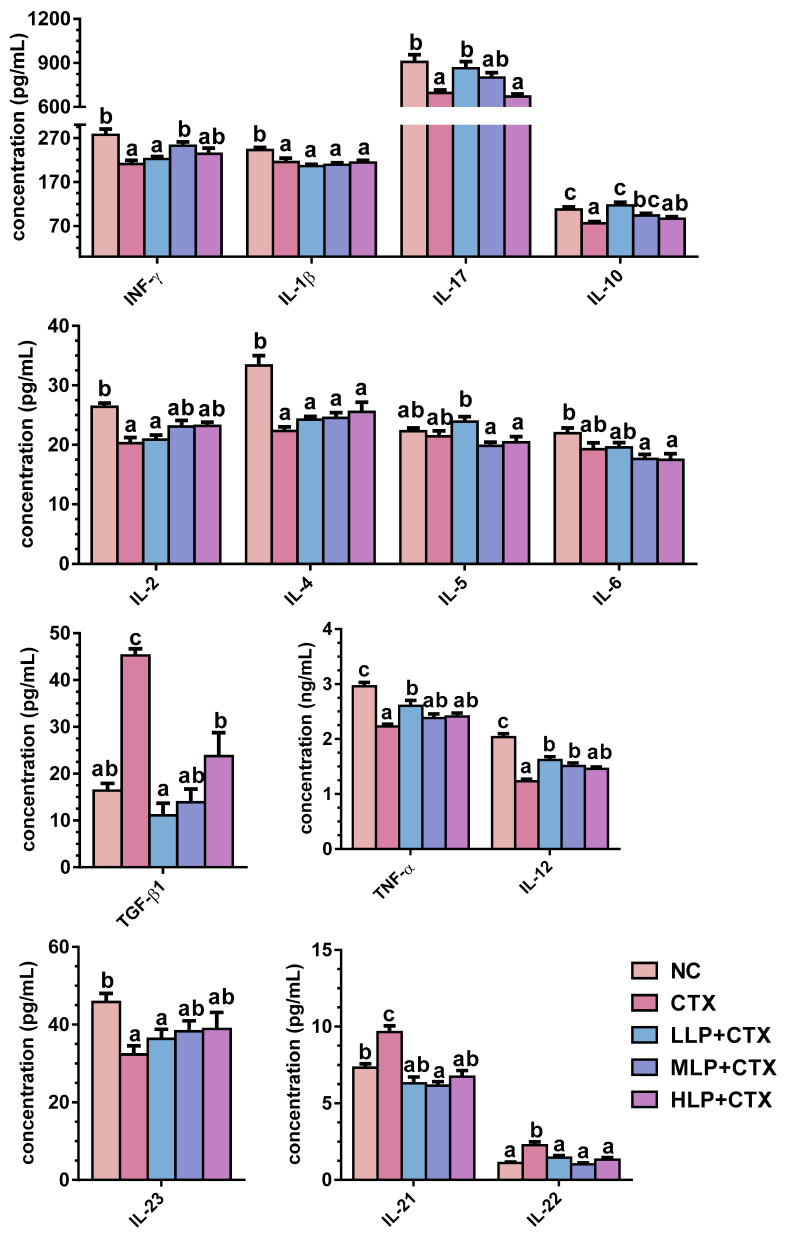
Effect of LP on serum cytokine concentration in CTX-treated mice. Error bars represent mean ± SEM (n = 10). Bars sharing different letters indicate *p* < 0.05 by *Tukey’s HSD*.

**Figure 2 foods-14-02901-f002:**
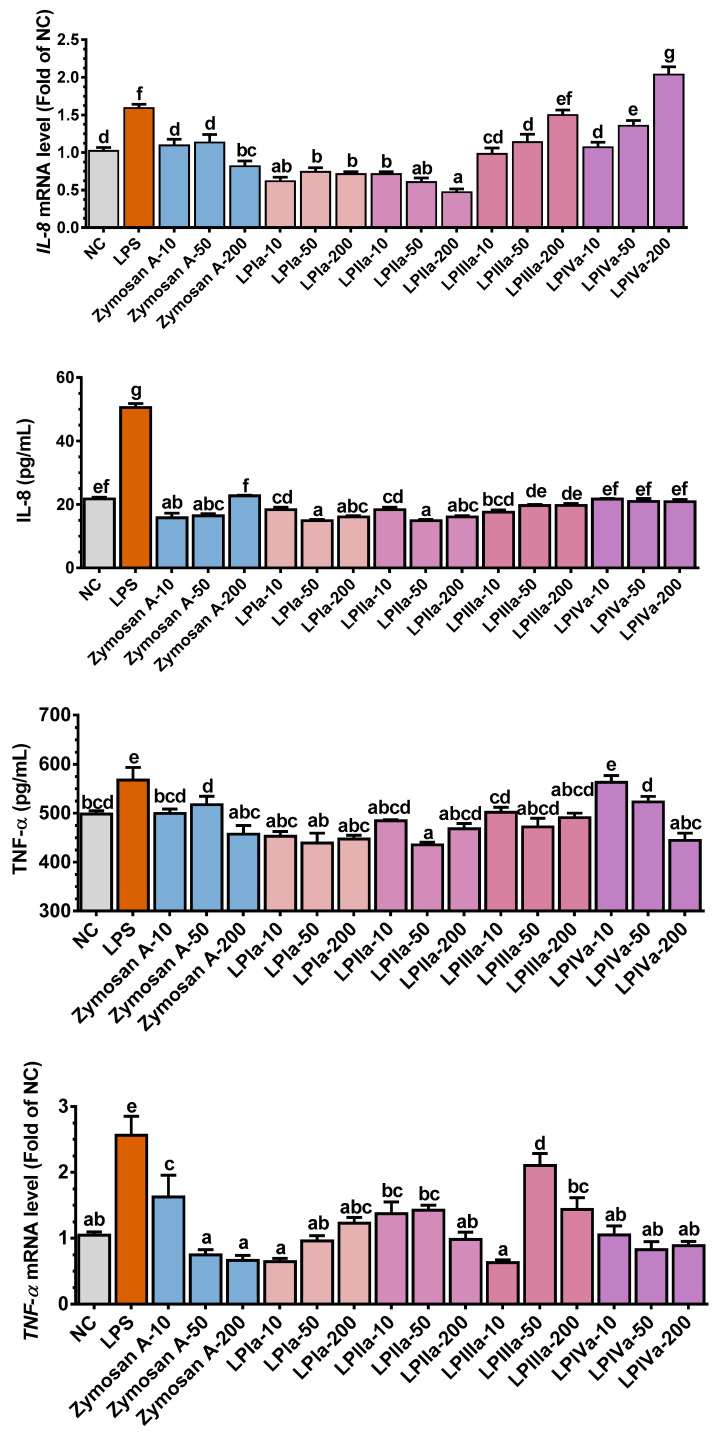
Effects of LP fractions on mRNA and protein expression of IL-8 in upper compartment and IL-6, TNF-α, and iNOS/NO in lower compartment in LPS-induced co-culture system of Caco-2 cells and RAW264.7 macrophages. Error bars represent mean ± SEM (n = 10). Bars sharing different letters indicate *p* < 0.05 by *Tukey’s HSD*.

**Figure 3 foods-14-02901-f003:**
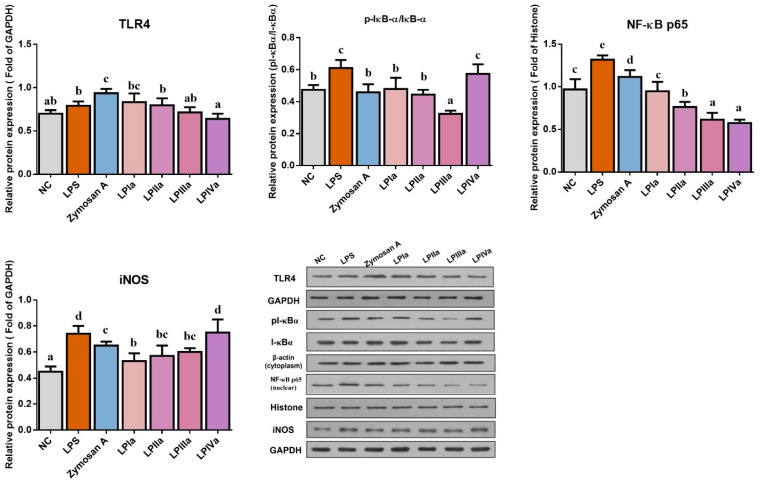
Effects of LP fractions on the protein expression of TLR4, p-IκB-α/IκB-α, NF-κB p65 (nuclear), and iNOS in LPS-induced co-culture Caco-2 cells and RAW264.7 macrophages. Error bars represent mean ± SEM (n = 3). Bars sharing different letters indicate *p* < 0.05 by *Tukey’s HSD*.

**Table 1 foods-14-02901-t001:** Effect of LP on serum immunoglobulin levels in CTX-treated mice.

Group	IgA (μg/mL)	IgG1 (μg/mL)	IgG2a (μg/mL)	IgG2b (μg/mL)	IgM (mg/mL)
NC	70.52 ± 0.87d	537.21 ± 33.93b	48.19 ± 3.00c	64.48 ± 3.34b	1.55 ± 0.02c
CTX	39.81 ± 1.21a	437.52 ± 14.50a	28.33 ± 1.41a	45.24 ± 3.38a	1.24 ± 0.05a
LLP + CTX	50.30 ± 1.19b	431.63 ± 27.66a	51.72 ± 1.70c	46.77 ± 2.21a	1.39 ± 0.05b
MLP + CTX	51.84 ± 1.99b	534.59 ± 9.83b	47.90 ± 2.41c	66.47 ± 3.99b	1.56 ± 0.05bc
HLP + CTX	57.92 ± 2.18c	462.76 ± 33.93ab	35.16 ± 1.09b	62.19 ± 2.67b	1.43 ± 0.02bc

Data represent mean ± SEM (n = 10). Mean values in the same column with different letters were significantly different (*p* < 0.05) by *Tukey’s HSD*.

**Table 2 foods-14-02901-t002:** Effect of LP on levels of SOD, MDA, GSH, and NO.

Group	SOD (U/mgprot)	MDA (nmol/mgprot)	GSH (μmol/gprot)	NO (μmol/gprot)
Liver				
NC	21.46 ± 0.30d	3.8 ± 0.13a	2.95 ± 0.17b	0.13 ± 0.01b
CTX	13.00 ± 1.01a	6.97 ± 0.52c	2.02 ± 0.13a	0.05 ± 0.01a
LLP + CTX	14.54 ± 0.43ab	5.17 ± 0.24b	2.68 ± 0.08b	0.06 ± 0.01a
MLP + CTX	16.02 ± 0.38bc	4.96 ± 0.17b	2.69 ± 0.13b	0.10 ± 0.01b
HLP + CTX	17.33 ± 0.22c	4.16 ± 0.19ab	2.94 ± 0.10b	0.13 ± 0.01b
Kidney				
NC	27.69 ± 0.17c	6.49 ± 0.07a	8.82 ± 0.27b	0.79 ± 0.05d
CTX	21.20 ± 0.93a	8.74 ± 0.17b	7.42 ± 0.27a	0.27 ± 0.02a
LLP + CTX	23.59 ± 0.68b	6.22 ± 0.28a	8.21 ± 0.22ab	0.42 ± 0.05b
MLP + CTX	26.50 ± 0.51c	6.33 ± 0.18a	8.57 ± 0.14b	0.56 ± 0.02c
HLP + CTX	29.07 ± 0.94c	6.73 ± 0.22a	8.10 ± 0.15ab	0.69 ± 0.03d
Intestine				
NC	36.11 ± 0.35c	0.71 ± 0.04a	119.30 ± 4.91c	0.82 ± 0.04d
CTX	25.84 ± 1.39a	0.96 ± 0.01c	77.35 ± 1.92a	0.24 ± 0.03a
LLP + CTX	27.73 ± 0.45ab	0.86 ± 0.02b	83.13 ± 2.89a	0.42 ± 0.04b
MLP + CTX	33.65 ± 1.20c	0.80 ± 0.02b	94.22 ± 3.00b	0.63 ± 0.02c
HLP + CTX	30.06 ± 0.65b	0.66 ± 0.01a	110.81 ± 3.69c	0.77 ± 0.02d
Serum	(U/mL)	(nmol/mL)	(μmol/mL)	(μmol/mL)
NC	93.63 ± 1.20c	6.86 ± 0.14a	32.97 ± 1.28c	6.16 ± 0.12d
CTX	80.96 ± 1.76a	10.28 ± 0.55b	20.86 ± 1.35a	2.08 ± 0.15a
LLP + CTX	87.62 ± 0.79b	8.23 ± 0.27a	22.71 ± 1.25a	2.91 ± 0.22b
MLP + CTX	97.16 ± 1.06cd	7.76 ± 0.16a	25.81 ± 0.85ab	5.42 ± 0.20c
HLP + CTX	100.01 ± 1.48d	8.08 ± 0.37a	30.30 ± 2.17bc	6.28 ± 0.09d

Data represent mean ± SEM (n = 10). Mean values of the same organ and different letters in the same column were significantly different (*p* < 0.05) by *Tukey’s HSD*.

## Data Availability

The original contributions presented in the study are included in the article/Appendix A, further inquiries can be directed to the corresponding authors.

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
