# Peer review of "Longan Polysaccharide as Adjuvant for Cyclophosphamide-Induced Side Effects in Murine Model"

_foods, 2025, doi:10.3390/foods14162901_

Round 1
Reviewer 1 Report
Comments and Suggestions for Authors
Manuscript Longan Polysaccharide as a Potential Adjuvant for Chemotherapy Side Effects: Immunomodulatory and Antioxidant Effects, ID: foods-3772608 describe an interesting study aimed to evaluate the prophylactic effect of Longan Polysaccharide (LP) on immune balance, inflammatory responses, and oxidative damage in CTX-induced immunosuppressed mice. In addition, using an in vitro model of intestinal epithelial cells, the authors stated to investigate the immunomodulatory mechanism of LP, to try to determine whether LP could serve as an effective adjuvant to alleviate the major side effects of chemotherapy drugs.
In following few lines I present the facts that led me to my final decision to advise the Authors to strengthen the current version of the work in accordance with the above proposals.
In the material and methods section, you refer to one of previous studies (reference 9). Although your work is identical in the mentioned part, it is certainly desirable that you also give a brief introduction to the methodology related to the extraction, purification and characterization of the fractions, with which you have also conducted the examination of the current work
Reading the paper, one gets the impression that it is a crude extract tested, and you are doing one part of the experiment with LP extract, but in fact, you are also using LP fractions specified as LP Ia - LP IVa. Therefore, additional more detailed explanation is needed, which opens up new questions.
In vivo experiments (3.1. LP Enhances Systemic Immunity of CTX-Treated Mice; 3.1.1. LP Mitigates Weight Loss in CTX-Treated Mice; 3.1.2. LP Promotes Serum Immunoglobulin Levels in CTX-Treated Mice; 3.1.3. LP Regulates Serum Cytokine Profiles in CTX-Treated Mice; 3.2. Antioxidant Effect of LP in CTX-Treated Mice) were performed using the whole extract (animals fed with LP extract, not with LP I a - LP IV a fractions)
In vitro tests described further (2.3. Intestinal Immunomodulatory Effect In Vitro and 3.3. Anti-Inflammatory Effect of Four LP Fractions in Caco-2/RAW264.7 Coculture Cells) were performed using fractionated LP extracts: LP Ia - LP IV a.
If you already had fractionated and structurally characterized extracts, why were your methodology and experiments not fully tested on fractions;
why the collective extract was not included in in vitro experiments elaborating 2.3. Intestinal Immunomodulatory Effect In Vitro and 3.3. Anti-Inflammatory Effect of Four LP Fractions in Caco-2/RAW264.7 Coculture Cells?
How is it possible to make a comparison based on a not a correctly designed experiment?
Methodology issue again, you have not explicitly defined the group you are referring to when saying:
"Cytokine and immunoglobulin concentrations in serum samples were tested by Bio-Plex Mouse (Group I, Group III)"
What does Group I, III is referred to?
Above all, burning issue is insufficient information provided when you refer to animal-related experiments primarily because no ethical clearance is supplied!
For papers of this type to be considered for publication, the status of the ethical approval of the study must be stated, along with the name of the Ethics Committee that approved the research study, as well as the date and number of the committee's approval. Only with the full indication of the required record number and date, that information can be considered complete!
Once these major details are addressed, I consider the work may be evaluated further.
Sincere greetings
Reviewer 2 Report
Comments and Suggestions for Authors
The paper meets the basic scientific and technical criteria. It needs some corrections.
When describing the examined plant in the introduction, it is necessary to provide more details, whether this plant is commercially available, whether it is grown as a plantation or is represented in natural conditions as a wild plant. At the end of the introduction, the idea and objectives should be better explained. Some Figures are of poor resolution, it is difficult to see the values. It is necessary to correct the figures for the purpose of better visibility. In the discussion, it is necessary to emphasize the scientific novelty, as well as the practical application of the results. In the references, it is necessary to italicize the Latin letters, as well as to check the italics in the text for plant names, because italics of plant names are necessary.
Reviewer 3 Report
Comments and Suggestions for Authors
The manuscript addresses a timely topic—using dietary polysaccharides to mitigate chemotherapy side effects—but requires clarifications in experimental design, deeper discussion of mechanisms, improved data presentation, and attention to writing clarity before further consideration.
Specific Comments:
Line 1–2: The title accurately conveys the study focus, but could be more informative by specifying the chemotherapy agent (cyclophosphamide) and model (murine) used.
Line 15–23 (Abstract): The abstract effectively summarizes findings but omits effect size for key immunoglobulin and cytokine changes. Suggest including quantitative fold-changes or percentage improvements for clarity.
Line 35–42 (Introduction): The authors should strengthen the literature review on existing adjuvants for chemotherapy side effects, highlighting gaps that longan polysaccharide (LP) specifically fills.
Line 49–54: The statement “intestinal immunity plays a pivotal role…” is broad. Recommend citing specific studies demonstrating LP’s effects on gut barrier integrity to justify the in vitro coculture model.
Line 58–60: The research question is implied but not explicit. Insert a clear hypothesis, i.e. (“We hypothesize that LP will attenuate CTX-induced immunosuppression via NF-κB modulation”).
Line 62–70 (Materials): Specify the purity and molecular weight range of the four LP fractions. Without these data, reproducibility is difficult.
Line 90–98 (Animal treatments): Clarify rationale for selecting LP doses (100, 200, 400 mg/kg). Were these based on prior toxicity or dose–response studies?
Line 109–116 (In vitro model): Provide TEER (transepithelial electrical resistance) values or microscopic confirmation that the Caco-2 monolayer integrity was intact prior to LPS exposure.
Line 125–128 (Statistical analysis): State whether data normality was assessed before ANOVA. Indicate software versions and post-hoc correction for multiple comparisons.
Table 1, Line 142–148: The use of superscript letters is appropriate, but the font size and legend placement render the table difficult to read. Recommend repositioning and enlarging the footnote.
Figure 1, Line 151–157: The multi-panel bar graphs lack error bars in some subplots. Ensure that SEM bars are shown consistently and described in the legend.
Line 160–167: The authors state that LP “ameliorated these CTX-induced changes in a dose-dependent manner,” but dose-response trends are not statistically tested (e.g., trend analysis). Include linear regression or equivalent.
Table 2, Line 170–172: Units for MDA (mmol/mgprot) appear unusually high. Verify unit accuracy—commonly it is nmol/mg protein—and correct if needed.
Line 174–181 (Results): The description of IL-6 and TNF-α changes in the coculture system would benefit from reporting both mRNA and protein correlation coefficients to support mechanistic claims.
Figure 2, Line 198–201: Panels A–H are crowded. Consider splitting into two figures or using supplemental materials. Also, label each panel clearly with corresponding LP fraction names.
Line 202–210: The statement “LPIVa > LPIIIa > LPIIa > LPIa” for TLR4 suppression should reference exact percentage inhibition values to substantiate ranking.
Line 220–227 (Discussion): The proposed mechanism involving acrolein binding to GSH merits citation of direct evidence for LP’s interaction with acrolein-derived adducts. Otherwise, it remains speculative.
Line 243–249: The manuscript suggests LP promotes B-cell proliferation and Th differentiation but provides no data. Rephrase as hypothesis or add supplemental flow cytometry on lymphocyte subsets.
Line 259–267: When comparing LP to other polysaccharides (e.g., Astragalus, lentinan), provide a concise table summarizing structural features and relative potencies to contextualize LP’s performance.
Line 287–295 (Conclusions): The conclusion emphasizes clinical promise. However, no safety/toxicity assessment is reported. Recommend tempering language or adding acute toxicity data.
References (Line 328–336): Several key recent studies on dietary polysaccharide adjuvants (2022–2025) are missing. Update literature to include latest clinical trials if available.
Writing quality: Instances of passive voice (“was conducted,” “were acquired”) hinder readability. Encourage active-voice revision for clarity.
Ethical considerations (Line 105–107): While animal ethics approval is noted, the text should specify the approval number and institution’s ethical guidelines reference to comply with journal standards.
Grammar/syntax: In Line 136–137, the phrase “mitigated CTX-caused weight loss, though post-CTX recovery did not differ significantly” is ambiguous—does LP affect only prophylaxis? Rephrase for precision.
Overall Summary:
The manuscript presents robust in vivo and in vitro data supporting longan polysaccharide’s immunomodulatory and antioxidant benefits against cyclophosphamide-induced toxicity. Strengths include comprehensive multi-organ analyses and mechanistic insights into NF-κB pathway modulation. However, critical details regarding LP fraction characterization, statistical analyses, figure clarity, and safety assessment are missing. Major revisions are needed to enhance reproducibility, data transparency, and alignment with clinical nutrition standards.
Reviewer 4 Report
Comments and Suggestions for Authors
The authors assessed the effect of the polysaccharides from longan, for their properties against chemotherapy side effects both in in vivo and in vitro models.
The study shows a noted effect of LP as a potential prophylactic in chemotherapy-induced side effects; however, some issues have been identified.
In introduction:
- The structural explanation of polysaccharides (branched neutral sugar chains, homogalacturonan backbones) is promising but needs clearer linkage to known bioactivities or prior findings in similar contexts (e.g., chemotherapy models).
- Reference [5] is mentioned regarding TLR4/NF-κB, but it is unclear whether this mechanism has been specifically demonstrated for LP or is a general claim.
- The statement that "systemic effects of LP on CTX-induced immunosuppression remain unclear" should be expanded with a concise summary of what is known (e.g., any prior in vivo or in vitro findings) and what is not known to justify the current study.
In methods:
- Although the authors cite the obtaining and preparation of the LP fractions, they should include a brief information about the raw material, how it was handled and what did they do to obtain such fractions. The key structural features (e.g. molecular weight, monosaccharide composition, purity, linkage types) or any type of characterization is essential for reproducibility.
- In the in vivo experiment, why did the authors use mentioned doses for the low, medium and high LP supplementation? Did they have any in-house results suggesting the use of such doses?
- The authors do not clarify how the co-culture was obtained in terms of days: differentiation od Caco-2 differentiation, confirmation of monolayer formation by TEER, and co-culture configuration (transwell setup). Also, the time points (2 h LP pre-treatment or LPS stimulation by 12 h) should be justified based on kinetics of immune activation or previous data.
- Any reference for the western blot assay?
Results
- Figure 1, 2 and 3: the size of letters is too small. It is quite difficult to understand what is shown. The authors should improve the visibility of the results. Also, they do not explain the different letters observed in the bars in the figure caption, along with the employed statistical analysis. In Figure 1, there are not subfigures but in Figure 2, the subfigures 2A, 2B, 2C, etc., are not explained in the figure caption.
- Using Student-Newman-Keuls is appropriate but somewhat outdated. Consider using Tukey’s HSD or Bonferroni correction if multiple comparisons were extensive. If the authors want to compare the data to a control, they can also consider the use of Dunnett’s post hoc test if appropriate.
- The results in TLR4/NF-kB pathway is informative but slightly repetitive. Consider restructuring it to show cause-effect more clearly.
Discussion
- Rephrase speculative statements (for instance, LP improves B cell/Th cell function) to be more tentative or support them with cited data.
- When citing studies on other polysaccharides, more clearly contrast their effects with LP to highlight novelty.
- Moderate strong claims about structure-activity relationships unless supported by specific data on LP (neutral sugar side chains are critical for TLR4 antagonism). The authors should show prove about this fact, even if it has been reported elsewhere.
Conclusions
- Condense repeated mechanistic details that were already discussed at length earlier and emphasize the main conclusion in a mor concise and impactful manner. Highlight what is novel about this study’s findings and how they advance current understanding or application of LP.
Minor:
Revise some typos in the manuscript. For instance, in line 44, resource should be ‘resources’. Or in line 93, five group should be five groups.

Round 2
Reviewer 1 Report
Comments and Suggestions for Authors
Dear Authors,
The works has been greatly improved by the addition of requested detailed explanations/comments.
Best regards
Reviewer 3 Report
Comments and Suggestions for Authors
While the authors demonstrated good faith efforts to address reviewer concerns, the remaining issues—particularly the lack of quantitative data in the abstract and unresolved statistical analysis concerns—require correction.
However, four critical deficiencies remain:
-
Abstract quantitative data: The abstract still omits specific fold-changes or percentage improvements for key immunoglobulin and cytokine findings. Revise the abstract to include quantitative changes (e.g., “1.83-fold increase in IgG2a”).
-
Title specificity: Update the title to “Longan Polysaccharide as Adjuvant for Cyclophosphamide-Induced Side Effects in Murine Model.
-
Dose-response analysis: Perform trend analysis for dose-response relationships or remove dose-dependence statements.
-
Figure quality: The crowding in Figure 2 persists; please reorganize or split the panels to enhance readability.
Additionally :
-
Explicitly acknowledge study limitations regarding mechanistic evidence.
- Strengthen the discussion of clinical translation challenges given the lack of toxicity data.
